# R-PTP-κ Inhibits Contact-Dependent Cell Growth by Suppressing E2F Activity

**DOI:** 10.3390/biomedicines10123199

**Published:** 2022-12-09

**Authors:** Hyun Ahm Sohn, Minho Kang, Hyunjung Ha, Young Il Yeom, Kyung Chan Park, Dong Chul Lee

**Affiliations:** 1Personalized Genomic Medicine Research Center, Korea Research Institute of Bioscience and Biotechnology (KRIBB), Daejeon 34141, Republic of Korea; 2Department of Biochemistry, School of Life Sciences, Chungbuk National University, Cheongju 28644, Republic of Korea; 3Department of Functional Genomics, University of Science and Technology (UST), Daejeon 34113, Republic of Korea

**Keywords:** R-PTP-κ, cell contact inhibition, E2F, p21, p27

## Abstract

Density-dependent regulation of cell growth is presumed to be caused by cell-cell contact, but the underlying molecular mechanism is not yet clearly defined. Here, we report that receptor-type protein tyrosine phosphatase-kappa (R-PTP-κ) is an important regulator of cell contact-dependent growth inhibition. R-PTP-κ expression increased in proportion to cell density. siRNA-mediated R-PTP-κ downregulation led to the loss of cell contact-mediated growth inhibition, whereas its upregulation reduced anchorage-independent cell growth in soft agar as well as tumor growth in nude mice. Expression profiling and luciferase reporter system-mediated signaling pathway analysis revealed that R-PTP-κ induced under cell contact conditions distinctly suppressed E2F activity. Among the structural domains of R-PTP-κ, the cytoplasmic domain containing the tandemly repeated PTP motif acts as a potent downregulator of the E2F pathway. Specifically, R-PTP-κ suppressed CDK2 activity through the induction of p21Cip1/WAF-1 and p27Kip1, resulting in cell cycle arrest at the G1 phase. In transcriptome-based public datasets generated from four different tumor types, R-PTP-κ expression was negatively correlated with the expression pattern and prognostic value of two known E2F1 target genes (*CCNE1* and *CDC25A*). Therefore, our results indicate that the R-PTP-κ-E2F axis plays a crucial role in cell growth-inhibitory signaling arising from cell-cell contact conditions.

## 1. Introduction

Cell growth is regulated by diverse extracellular signals including biological, chemical, and physical actions [1]. In multicellular organisms, physical contact between neighboring cells under confluent growth conditions can act as an important physiological signal that reversibly regulates cell growth in a density-dependent manner [2]. The corresponding cellular response, contact inhibition, is a unique program of cellular growth inhibition that occurs when cells make contact with one another at a high density [3,4,5]. It is well established that contact inhibition arrests cells in the G0-G1 phase of the cell cycle in association with an increase in the expression of the cyclin-dependent kinase inhibitor p27Kip1 (p27) [6,7]. Growth signals induced by the epidermal growth factor (EGF) or basic fibroblast growth factor (b-FGF) are also suppressed under cell-cell contact conditions [8]. Loss of contact inhibition is associated with accelerated cell growth, and prolonged loss of contact inhibition can cause abnormal growth states, such as malignant transformation and cancer [9,10]. Despite the physiological and pathological importance of contact inhibition, the salient features of the underlying molecular mechanisms and signal transduction pathways are poorly understood.

Regulation of tyrosine phosphorylation by protein tyrosine kinases (PTKs) and phosphatases (PTPs) is critical for controlling cell growth in response to extracellular signals. Unlike many PTKs that function as oncoproteins, loss of function is frequently associated with tumorigenesis of PTPs in different cancer types [11]. PTPs are structurally classified into two distinct groups depending on the presence of transmembrane domains: receptor-like PTPs and non-transmembrane PTPs [12]. The expression of receptor-like PTPs is upregulated in association with cell contact inhibition [13,14,15,16]. It has been postulated that contact-dependent regulation requires cell surface receptors engaged in the physical interaction between cell surfaces to initiate contact inhibition [5]. Notably, many subfamilies of receptor-like PTPs (R-PTPs) have extracellular segments with features of cell adhesion molecules that are implicated in processes involving cell–cell and cell-matrix contact [12]. Some members of these subfamilies have been shown to be upregulated in association with cell contact inhibition [13,14,15,16].

The receptor-type protein tyrosine phosphatase-kappa (R-PTP-κ) contains several domains, including tandem intracellular catalytic domains, a transmembrane domain, and a large extracellular region consisting of an MAM (meprin/A5/PTPmu) domain, an immunoglobulin-like domain, and four fibronectin type-III repeats [17]. Various motifs in the extracellular domain are found in many adhesion molecules and can induce homophilic intercellular interactions [18,19]. R-PTP-κ protein is localized to cell-cell contact sites in association with the cadherin-catenin complex in epithelial cell lines and organs (e.g., pancreas), suggesting that it may play important roles in the processes involving cell-cell contact [20,21,22]. Intracellular PTP domains can catalyze the dephosphorylation of β-catenin and epidermal growth factor receptor (EGFR) [20,23], and are implicated in the function of R-PTP-κ as a tumor suppressor gene. However, the precise roles and molecular mechanisms of R-PTP-κ in density-dependent regulation of cell growth remain unclear.

Here, we describe the cellular mechanisms that govern density-dependent growth arrest mediated by R-PTP-κ. We demonstrated that R-PTP-κ is increasingly expressed under high cell density conditions that promote cell-cell contacts and induce inhibition of the E2F pathway activity, resulting in cell cycle arrest at the G1 phase. Our results suggest that R-PTP-κ is a crucial factor in the regulation of cell contact-mediated growth responses.

## 2. Materials and Methods

### 2.1. Reagents

Antibodies for R-PTP-κ, β-actin, p21^Cip1/Waf−1^ (p21), p27^Kip1^ (p27), and CDK2 were purchased from Santa Cruz Biotechnology (Santa Cruz, CA, USA). The primers used for mRNA detection were as follows: R-PTP-κ (forward), GCTGCTCTTATGGACAGC TACAG and R-PTP-κ (reverse), GATTCCAGGTACTCCAAAGCTACA; β-actin (forward), CTGGAGAAGAGCTACGAGCTGC and β-actin (reverse), CTAGAAGCATTTGCGG TGGACG. Synthetic R-PTP-κ siRNAs were purchased from Samchully Pharm (siRNA-1 and siRNA-2; Seoul, South Korea), and the sequences were as follows: siRNA-1 (forward): AACCAUCUGCCACCUUAUACAAAdTdG and siRNA-1 (reverse), CAUUUGUAUAA GGUGGCAGAUGGUUCA; siRNA-2 (forward), AGAUUAGUGUAUGAUUAUGGCUG dTdA and siRNA-2 (reverse), UACAGCCAUAAUCAUACACUAAUCUCC; GFP (forward), GUUCAGCGUGUCCGGCGAGTT and GFP (reverse), CUCGCCGGACACG CUGAACTT.

### 2.2. Cell Culture and Reporter Assay

Chang Liver, HeLa, and normal human fibroblast (HFT) cells were maintained in Dulbecco’s Modified Eagle Medium (DMEM) supplemented with 10% fetal bovine serum (Hyclone, Logan, UT, USA). For reporter assays, cells were seeded at a density of 2 × 10^5^ cells/well in 6-well plates, incubated for 24 h, and then transfected with 1 µg DNA/well using Lipofectamine Plus^TM^ reagent (Invitrogen, Waltham, MA, USA), according to the supplier’s protocol. After transfection, the cells were incubated for an additional 48 h and harvested for luciferase assay. The cell pellet was disrupted in 100 µL reporter lysis buffer (Promega, Madosin, WI, USA), and luciferase activity was measured using the Promega Luciferase Assay System and Lumat LB 9501 Single Tube Luminometer (Berthold, Bad Wildbad, Germany). Renilla luciferase was used as an internal control to normalize transfection efficiency. Luciferase assays were performed in duplicate and each experiment was repeated at least twice.

### 2.3. Proliferation Assay

Cell growth was determined based on the incorporation of [^3^H]-thymidine. Density-dependent effects were measured by plating the cells at low (1 × 10^4^ or 5 × 10^3^ cells/well in 6- and 12-well plates, respectively), medium (5 × 10^4^ or 1 × 10^4^ cells/well in 6- and 12-well plates, respectively), and high densities (4 × 10^5^ or 1 × 10^5^ cells/well in 6- and 12-well plates, respectively), cultured for 48 h, and then incubated for an additional 6 h with [^3^H]-thymidine (0.2 μCi/well, Amersham). Cells were washed twice with ice-cold PBS, and unincorporated [^3^H]-thymidine was removed by washing with ice-cold 10% trichloroacetic acid (TCA; Sigma-Aldrich, St. Louis, MO, USA) for 5 min. Cells were dissolved in 10% SDS at room temperature, and the amount of [^3^H]-thymidine uptake was determined using liquid scintillation spectrometry (LS 5801, Beckman, Albertville, MN, USA). Values are expressed as mean cpm of duplicate samples.

### 2.4. In Vitro Kinase Assay

Protein lysates (200 μg) in lysis buffer (1% Triton X-100, 150 mM NaCl, 100 mM KCl, 20 mM HEPES pH 7.9, 10 mM EDTA, 1 mM sodium orthovanadate [SOV], 10 μg/mL aprotinin, 10 μg/mL leupeptin, and 1 μM PMSF) were incubated with anti-Cdk2 antibody (1 μg/mL) overnight at 4 °C. Following this, 20 μL Protein G-agarose (Santa Cruz Biotechnology) was added, and the lysates were incubated for 2 h at 4 °C. Protein G-agarose conjugates were pelleted by centrifugation, washed thrice with lysis buffer and twice with kinase buffer (50 mM Tris pH 7.5, 10 mM MgCl_2_, 1 mM DTT, 1 μM PMSF, 10 μg/mL leupeptin, and 10 μg/mL aprotinin), and then resuspended in 30 μL kinase buffer. Samples were incubated at 37 °C for 30 min with 20 μM ATP, 5 μCi [γ-32P]-ATP (Amersham), and 1 μg histone H1 (US Biological, Salem, MA, USA). The reaction was stopped by adding 5X sample loading buffer (60 mM Tris, pH 6.8, 25% glycerol, 2% SDS, 14.4 mM β-mercaptoethanol, and 0.1% bromophenol blue). The kinase reaction mixtures were separated by SDS-PAGE on 12% gels, and phosphorylated substrates were detected by autoradiography.

### 2.5. Assay of Anchorage-Independent Cell Growth

The clonogenic assay was performed as described previously [10] using Chang Liver cells overexpressing R-PTP-κ and Hep3B cells in which R-PTP-κ expression was silenced by stable expression of R-PTP-κ-targeted shRNA purchased from Samchully Pharm. Triplicate cell samples (1 × 10^4^ cells/well) were seeded in 6-well plates (Corning Costar, Bodenheim, Germany). After incubation for 2–3 weeks, the colonies were counted and photographed.

### 2.6. Tumorigenesis Assay in Nude Mice

Eight-week-old female BALB/c-nu mice were purchased from Central Animal Laboratory (SLC, Shizuoka, Japan). Mock-transfected or R-PTP-κ-overexpressing Chang Liver cells (3 × 10^6^ cells/mL) were injected into the flanks of nude mice (n = 5 mice/sample). Tumor growth was measured with calipers at the indicated time points, starting on the third week after injection. Tumor volumes were calculated using the length (a), width (b), and height (c) measurements ([a × b × c]/2). All procedures involving animal experiments were approved by the Animal Research Ethics Committee of Korea Research Institute of Bioscience and Biotechnology.

### 2.7. Statistical Analysis

All statistical data are presented as mean ± standard deviation (SD). The *p* values for determining statistical significance were calculated using an unpaired two-tailed Student’s *t*-test. The symbols used are: *, *p* < 0.05; **, *p* < 0.01.

## 3. Results

### 3.1. Induction of R-PTP-κ Expression Is Crucial to Cell Contact-Mediated Growth Inhibition

To investigate whether the induction of R-PTP-κ is necessary for the generation and maintenance of growth inhibitory signals under cell-contact conditions, we tested the effects of plating different cell densities on R-PTP-κ mRNA expression and the growth of HeLa and HFT cells. Cell proliferation rates were determined by measuring [^3^H]-thymidine incorporation at three different cell densities. As shown in Figure 1A, the growth rate was closely correlated with the R-PTP-κ mRNA expression level, which increased in proportion to the cell plating density. Using confocal immunofluorescence microscopy, we found that endogenous R-PTP-κ protein induced in HeLa cells under cell contact conditions was localized to the plasma membrane (Figure 1B). To confirm the specificity of the fluorescent signal, we transiently transfected cells with an R-PTP-κ expression vector (R-PTP-κ^OE^) and observed much stronger fluorescence in the plasma membrane. We then measured the density-dependent growth rates of HeLa cells in which R-PTP-κ expression was knocked down by R-PTP-κ siRNA (designated κ-siRNA). The κ-siRNA–mediated depletion of R-PTP-κ at different cell densities was confirmed by RT-PCR (Figure 1C). In R-PTP-κ-silenced HeLa cells, [^3^H]-thymidine incorporation increased in proportion to increasing plating density, whereas it was virtually unchanged regardless of the plating density in control cells (Figure 1D). These results indicate that cell contact-dependent growth inhibition requires induction of R-PTP-κ. Next, to determine whether the role of R-PTP-κ in the control of cell growth had notable implications in tumorigenesis, we examined its effect on anchorage-independent cell growth in soft agar (Figure 1E). We found that colony formation in Chang Liver cells stably overexpressing R-PTP-κ was significantly reduced compared to mock-transfected cells. Conversely, the shRNA-mediated knockdown of R-PTP-κ in Hep3B cells, in which endogenous R-PTP-κ level is relatively high, resulted in enhanced colony formation. We further examined the effect of R-PTP-κ on the regulation of tumorigenesis in vivo using a mouse xenograft model. Nude mice were transplanted with Chang Liver cells stably overexpressing R-PTP-κ and evaluated for tumor growth (Figure 1F). The volume of tumors formed by R-PTP-κ-expressing cells was significantly smaller than those formed by mock-transfected control cells (54.3%, *p* < 0.001). Collectively, these results indicate that R-PTP-κ suppresses tumor cell features, including loss of cell contact inhibition, anchorage-independent cell growth in soft agar, and tumor growth in nude mice, suggesting that R-PTP-κ may act as a tumor suppressor in human carcinogenesis.

### 3.2. R-PTP-κ Negatively Regulates EGFR Pathway under Cell Contact Inhibition

To determine whether cell growth inhibition under cell contact conditions is related to the activity of R-PTP-κ, we analyzed global protein tyrosine phosphorylation status according to R-PTP-κ expression changes. Ectopic expression of R-PTP-κ effectively suppressed protein tyrosine phosphorylation levels in HeLa cells (Figure 2A, left) as expected. Next, we assessed the effect of R-PTP-κ depletion on the protein tyrosine phosphorylation patterns in EGF stimulated cells grown at different densities. In control cells, there was a significant decrease in overall phosphotyrosine levels with an increase in cell density, but in κ-siRNA-transfected cells, this density-dependent difference was minimal (Figure 2A, right). R-PTP-κ has been shown to dephosphorylate EGFR Tyr1068 and Tyr1173 residues [23], which are important for the EGF mitogenic signaling. We confirmed that overexpression of R-PTP-κ results in dephosphorylation of EGFR at Tyr-1173 (Figure 2B), and importantly, the p-Tyr-1173-EGFR level was inversely correlated with the cell density, and substantially increased upon depletion of R-PTP-κ expression (Figure 2C). These results demonstrate that R-PTP-κ blocks mitogenic signaling in a cell density-dependent manner.

### 3.3. R-PTP-κ Negatively Regulates E2F Pathway under Cell Contact Inhibition

Gene expression profiling is a useful tool for predicting and investigating the mechanisms of various biological processes. Thus, to identify novel signaling pathways related to R-PTP-κ-mediated cell-contact inhibition, we performed RNA sequencing in HeLa cells treated with a tyrosine phosphatase inhibitor, SOV, or κ-siRNA at different cell densities. Next, we calculated the enrichment score using 50 hallmark gene sets derived from BROAD [httpa://www.gsea-msigdb.org (21 January 2021)]. Among the various cancer hallmark-based pathways, E2F, NF-kB, and Myc pathway activities scored low under cell contact conditions, but their activity was high under κ-siRNA or OVA treatment conditions (Figure 3A). These data indicate that R-PTP-κ-mediated cell contact inhibition may be associated with cell growth-related pathways, including E2F signaling. To test this hypothesis, we analyzed the activity of growth-related biochemical pathways using a luciferase reporter system at different cell densities. As shown in Figure 3B, the activity of many pathways increased in proportion to cell plating density. However, the activity of the E2F- and NF-κB-responsive reporters was suppressed at high cell density compared to that at medium cell density. Consistent with the results of the gene expression profiling, E2F- or NF-κB-mediated cell signaling may be closely involved in the regulation of density-dependent cell growth. We then assessed changes in the activity of these reporters upon R-PTP-κ silencing in HeLa cells to determine the pathway directly responsible for R-PTP-κ-mediated cell contact-dependent growth signaling. R-PTP-κ depletion resulted in an approximately 2.2-fold increase in E2F-dependent luciferase activity, without a notable change in NF-κB activity (Figure 3C). The E2F-dependent luciferase reporter values decreased at a higher plating density in control HeLa cells but increased with the increase in plating density in R-PTP-κ-silenced cells (Figure 3D). These results strongly suggest that contact-dependent cell growth inhibition is intimately associated with the suppression of E2F activity by R-PTP-κ. Because the E2F pathway is involved in cell cycle progression, we compared the cell cycle pattern of κ-siRNA-transfected cells with that of control cells using FACS analysis (Figure 3E). In control HeLa cells grown at two different plating densities, the G1 fraction increased in the higher density group compared to the lower density group, while the S and G2 fractions decreased. However, R-PTP-κ-depleted cells exhibited the opposite cell cycle pattern, with a decreased G1 fraction and increased S and G2 fractions under high cell density conditions. Moreover, HeLa cells treated with SOV showed decreased G1 and increased S and G2 fractions at high cell density. These results suggest that PTP activities are crucial in inhibiting cell cycle progression, and that density-dependent induction of R-PTP-κ expression may play a major role in mediating cell contact-dependent inhibition of cell cycle progression.

### 3.4. Molecular Mechanism for the Repression of E2F Pathway by R-PTP-κ

Inhibition of cyclin-dependent kinases (cdks) causes the accumulation of under-phosphorylated forms of retinoblastoma (RB) protein, which bind E2Fs and render them transcriptionally inactive [24]. To investigate the molecular mechanism underlying R-PTP-κ-mediated suppression of E2F activity, we examined the expression pattern of the cdk inhibitors (CKIs) p21 and p27 in response to changes in R-PTP-κ levels. p21 protein levels in κ-siRNA-transfected cells were much lower than those in control cells, especially in the high-density plating group (Figure 4A). Similar effects were observed for p27 levels. These results indicate that R-PTP-κ is required for the cell density-dependent upregulation of p21 and p27, which, in turn, repress E2F activity. We examined the mechanism of R-PTP-κ-dependent upregulation of p21 and p27 expression by measuring their promoter activity in the presence of different levels of R-PTP-κ in HeLa cells. The activity of the p21 promoter was significantly increased by R-PTP-κ overexpression but decreased by R-PTP-κ depletion with siRNA (Figure 4B). We also found that the R- PTP-κ-induced upregulation of p21 transcription was not mediated by p53, since neither the p53 protein level nor the activity of the p53-responsive reporter changed in response to changes in the R-PTP-κ level (Figure 3C). In contrast, the activity of the p27 promoter remained unchanged by overexpression or siRNA-mediated knockdown of R-PTP-κ (Figure 4B, right panel). It has previously been reported that p27 protein levels are mainly regulated by ubiquitin-dependent proteolysis [25]. Both p21 and p27 negatively regulate the activity of CDK2, which is necessary for cell cycle progression through the G1 phase. Therefore, we examined whether R-PTP-κ negatively regulated CDK2 activity. Using an in vitro kinase activity assay, we found that CDK2 activity was clearly reduced in R-PTP-κ-overexpressing cells but was elevated in R-PTP-κ-depleted cells (Figure 4C). These results indicate that R-PTP-κ negatively regulates the cell cycle by suppressing CDK2 activity via the upregulation of p21 and p27.

Next, we determined the functional domains of R-PTP-κ that mediate contact-dependent growth inhibition by measuring the effects of R-PTP-κ structural domains on the activity of the E2F pathway in HeLa cells. Three extracellular domains (MAM, Ig, and FN) and a cytoplasmic domain containing tandemly repeated PTP motifs were cloned into a FLAG-tagged eukaryotic expression vector (Figure 4D, upper panel). Their expression was confirmed using western blot analysis. First, we investigated whether the extracellular domains affected E2F activity when acting as a ligand by expressing the soluble form. However, these three extracellular domains did not affect E2F activity. Interestingly, in contrast to the extracellular domains, the expression of the isolated PTP domain significantly downregulated E2F activity in HeLa cells (Figure 4D, lower panel). We then examined the relevance of the PTP domain of R-PTP-κ in the regulation of E2F activity by expressing the isolated PTP domain with concomitant depletion of endogenous R-PTP-κ expression. Introduction of an siRNA specific to endogenous R-PTP-κ (siRNA-1 in Figure 4E) into HeLa cells expressing the isolated PTP domain failed to alleviate PTP domain-mediated downregulation of E2F activity. However, another siRNA that knocked down endogenous R-PTP-κ and exogenously provided the PTP domain (siRNA-2 in Figure 4E) abrogated the effect of the isolated PTP domain, restoring E2F activity to normal levels (Figure 4E). These results suggest that the PTP domain alone can act as a strong downregulator of the E2F pathway independent of the extracellular domain.

### 3.5. R-PTP-κ Is Negatively Correlated with E2F1 Target Genes in Various Tumor Tissues

To assess the possible clinical implications of R-PTP-κ by evaluating its relationship with E2F activity in patient tumor tissues, we analyzed the expression patterns of R-PTP-κ and known E2F1 target genes in public datasets generated from kidney renal clear cell carcinoma (KIRC, n = 534), colon and rectal adenocarcinoma (COAD, n = 469), cervical cell carcinoma (KIRC, n = 534), colon and rectal adenocarcinoma (COAD, n = 469), cervical squamous cell carcinoma and endocervical adenocarcinoma (CESC, n = 304), and liver hepatocellular carcinoma (LIHC, n = 370). Patients were divided into R-PTP-κ high and low expression groups, and the expression correlation between R-PTP-κ and two E2F1 target genes (*CCNE1* and *CDC25A*) related to cell cycle regulation was examined. As shown in Figure 5A, *CCNE1* expression was lower in the patient group with high levels of R-PTP-κ than in the R-PTP-κ low expression group, and its expression was inversely correlated with R-PTP-κ expression in the KIRC (*p* = 7.43 × 10^−23^), COAD (*p* = 3.63 × 10^−6^), CESC (*p* = 0.01523), and LIHC (*p* = 0.10184) cohorts. The expression level of *CDC25A*, another E2F1 target gene, was also lower in the R-PTP-κ-high group than in the R-PTP-κ-low group (Figure 5B). The reduction of these genes in the R-PTP-κ-high group indicates that R-PTP-κ negatively regulates the expression of E2F1 target genes *CCNE1* and *CDC25A* in various tumor tissues. We then analyzed the overall survival according to the expression pattern of R-PTP-κ or E2F1 target genes in the KIRC and LIHC cohorts. Patients with high R-PTP-κ expression had a significantly better prognosis than those with low R-PTP-κ expression (Figure 5C,D). However, in the same cohort, patients with high *CCNE1* or *CDC25A* expression were classified into the poor prognosis group, showing an opposite pattern to that of R-PTP-κ. These results suggest that the inverse relationship between R-PTP-κ and the two genes may be related to R-PTP-κ-mediated regulation of E2F1 activity.

## 4. Discussion

We identified the role of R-PTP-κ in controlling cell growth in the context of cell-cell contact. R-PTP-κ expression was upregulated in proportion to the density of cells in the culture, and the induction pattern paralleled the growth inhibition caused by density-dependent cell contacts. In addition, R-PTP-κ counteracted mitogenic signals by effectively inhibiting E2F activity, causing cell cycle arrest at the G_1_ phase. Therefore, we suggest that R-PTP-κ is a critical regulator of cell-cycle progression in the context of changes in cell-cell contact events.

By examining the density-dependent changes in the activity of representative growth-related biochemical pathways using luciferase reporters (Figure 3B), it was found that E2F pathway activity was decreased under high cell density conditions in an R-PTP-κ-dependent manner. Interestingly, in contrast to the suppression of E2F activity in parallel with the pattern of cell contact-mediated growth inhibition in the control cells, the activities in κ-siRNA-treated cells were increased in proportion to the cell number irrespective of cell density (Figure 3D). The relationship between E2F activity and cell contact-mediated growth inhibition has been reported by several researchers [26,27]. Therefore, the regulation of E2F activity may be important for the inhibition of cell contact-induced growth. A phenomenon caused by cell-cell contact is an increase in CDK inhibitors. It is well known that the CKI proteins p21 and p27 are negative regulators of E2F-driven cell-cycle progression and markers of cell contact inhibition, which can also be induced by TGF-β1 treatment under cell contact conditions [7,28]. In the present study, we also showed that the expression of p21 and p27 proteins, which increased at a high cell density, significantly decreased upon R-PTP-κ silencing (Figure 4A). Consistently, R-PTP-κ negatively regulated the activity of CDK2, causing cell cycle arrest at the G1/S checkpoint. These results suggest that R-PTP-κ-mediated growth inhibition is tightly linked to suppression of the E2F pathway via regulation of p21 and p27 protein expression.

R-PTP-κ belongs to the type II receptor-like PTP family, carrying tandem intracellular catalytic domains and a large extracellular region [12]. Among the distinct motifs constituting the extracellular segment, the MAM (meprin/A5/PTPmu) domain is necessary for homophilic interactions with neighboring cells [29]. It has recently been reported that R-PTP-κ expression increases by homophilic interactions under cell-contact conditions [30]. Interactome and quantitative tyrosine phosphoproteomics studies have shown that the intracellular PTP domain of R-PTP-κ interacts with cell junction-related proteins and directly dephosphorylates their proteins to promote cell-cell adhesion. Thus, the PTP domain with cellular phosphatase activity can effectively modulate cell behaviors, such as cell adhesion, in response to external stimuli. Another study that R-PTP-κ dephosphorylated p-Tyr1068 and p-Tyr1173 of EGFR of which the phosphorylation was important for mitogenic signaling of EGF. Our finding also revealed minor decrease in the intensity of p-Tyr1173-EGFR at high cell density when treated with κ-siRNA (Figure 2C). These finding are consistent with previous results and indicates that R-PTP-κ acts as a regulator for removing EGFR mitotic signals at high cell densities. Interestingly, in our study of the role of individual domains in R-PTP-κ-mediated cell growth regulation, extracellular motifs did not affect E2F activity, but the intracellular PTP domain acted as a strong negative regulator of the E2F pathway (Figure 4D). Based on the above results, it can be envisaged that during cell contact, R-PTP-κ is upregulated by homophilic interactions through the extracellular region and triggers a negative growth signal via PTP activity.

According to previous reports, loss of cell contact inhibition is associated with tumor cell transformation. We have shown that overexpression of R-PTP-κ significantly suppressed colony formation in Chang Liver cells which have low endogenous level of R-PTP-κ, and κ-siRNA enhanced the colony formation in Hep3B cells which have relatively high endogenous level of R-PTP-κ (Figure 1E). Therefore, the expression level of R-PTP-κ was implicated in the anchorage-independent growth of cells in soft agar, indicating that R-PTP-κ may play an important role in the regulation of human tumorigenesis. In addition to the literature, this provides further evidence to suggest that R-PTP-κ is a good candidate tumor-suppressor gene; it is upregulated by TGF-β1 and maps to 6q22.2–22.3, which is frequently deleted in melanomas [31,32]. Furthermore, loss of R-PTP-κ expression has been observed advanced breast cancer and in 76% of primary central nervous system lymphomas (PCNLs) [33,34]. R-PTP-κ is a tumor suppressor that dephosphorylates and inactivates oncogenic proteins such as STAT3, EGFR and CD133 [35,36,37]. Chang Liver cells stably expressing R-PTP-κ had smaller tumors than control cells in nude mice, suggesting that R-PTP-κ may act as a putative tumor suppressor (Figure 1F). The prognostic values of R-PTP-κ in the published KIRC and LIHC cohorts showed that high levels of R-PTP-κ expression were significantly associated with a good prognosis (Figure 5C,D). In addition, R-PTP-κ expression was negatively correlated with the expression patterns and prognostic value of two known E2F1 target genes (*CCNE1* and *CDC25A*), suggesting that R-PTP-κ may be involved in signal transduction pathways that antagonize E2F1-mediated tumor progression in various tumor tissues.

## 5. Conclusions

Cell contact-induced R-PTP-κ has been implicated in the suppression of E2F activity. R-PTP-κ negatively regulates the activity of CDK2 kinase, which controls cell cycle progression. The results presented herein demonstrate the molecular mechanism by which R-PTP-κ inhibits cell contact-mediated growth. However, it is yet to be elucidated whether there are other substrate(s) of R-PTP-κ that suppress E2F activity.

## Figures and Tables

**Figure 1 biomedicines-10-03199-f001:**
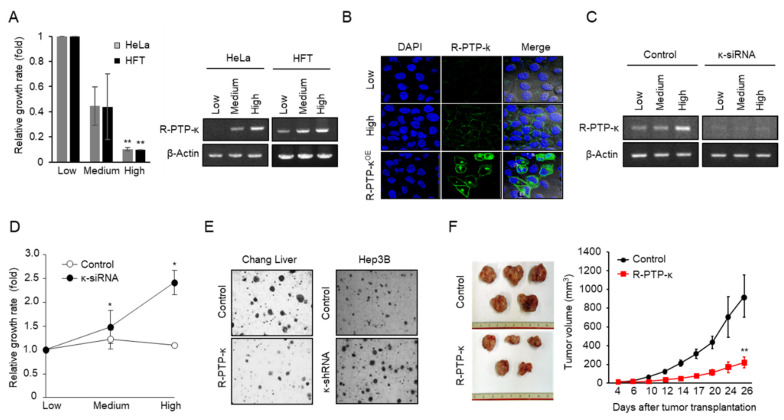
Effect of R-PTP-κ on density-dependent cell growth. (**A**) Cells were plated at three different densities and their growth rates were measured. The relative growth rates were normalized by the number of initial cell seeding. Cell density-dependent changes in R-PTP-κ mRNA expression were analyzed using RT-PCR (right panel). ** *p* < 0.01. (**B**) Induction of R-PTP-κ protein in HeLa cells under cell-contact conditions was analyzed using immunofluorescence microscopy. (**C**) Validation of optimal siRNA for RNAi-mediated R-PTP-κ silencing. R-PTP-κ expression was analyzed using RT-PCR. (**D**) Loss of cell-contact inhibition due to R-PTP-κ knockdown. HeLa cells transfected with the control or κ-siRNA were plated at three different densities and their growth rates were measured. * *p* < 0.05. (**E**) R-PTP-κ suppresses anchorage-independent cell growth. Clonogenicity in soft agar was measured using Chang Liver cells overexpressing R-PTP-κ and Hep3B cells expressing κ-siRNA. Microscopic images were taken at 10X magnification. (**F**) R-PTP-κ represses tumor formation in nude mice. Each mouse was subcutaneously injected with Chang Liver cells (3 × 10^6^ cells/mL). Tumor volumes were measured and compared between two groups using Student’s *t*-test. ** *p* < 0.01.

**Figure 2 biomedicines-10-03199-f002:**
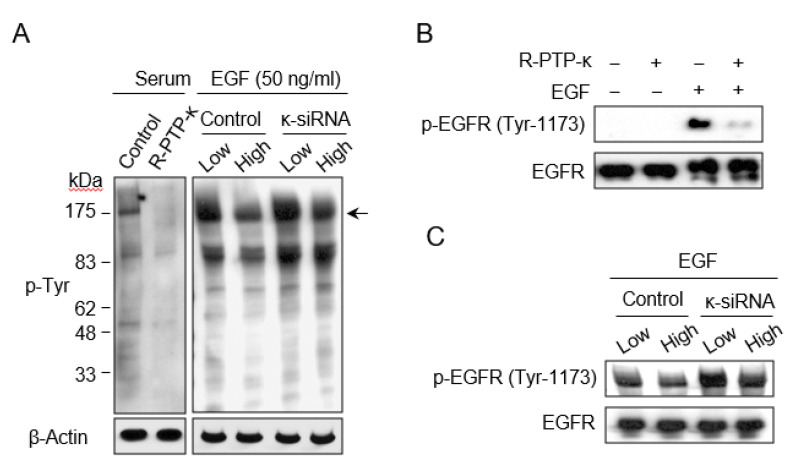
The status of global protein tyrosine phosphorylation status in relation to cell density and R-PTP-κ expression level. (**A**) Effect of R-PTP-κ on the regulation of EGFR phosphorylation. Serum-starved HeLa cells were stimulated with 10% FBS (left panel) or 50 ng/mL of EGF (right panel) for 10 min, and the cell lysates were analyzed using immunoblotting with anti-p-Tyr antibody. β-Actin was used as the loading control. Arrow indicates the position of 170-kD band. (**B**) Regulation of EGF-induced tyrosine phosphorylation of EGFR by R-PTP-κ overexpression. (**C**) Regulation of EGF-induced tyrosine phosphorylation of EGFR by R-PTP-κ knock-down. EGF-induced tyrosine phosphorylation of EGFR was analyzed using Western blotting with anti-p-EGFR (Tyr-1173) antibody. Total EGFR was used as the loading control.

**Figure 3 biomedicines-10-03199-f003:**
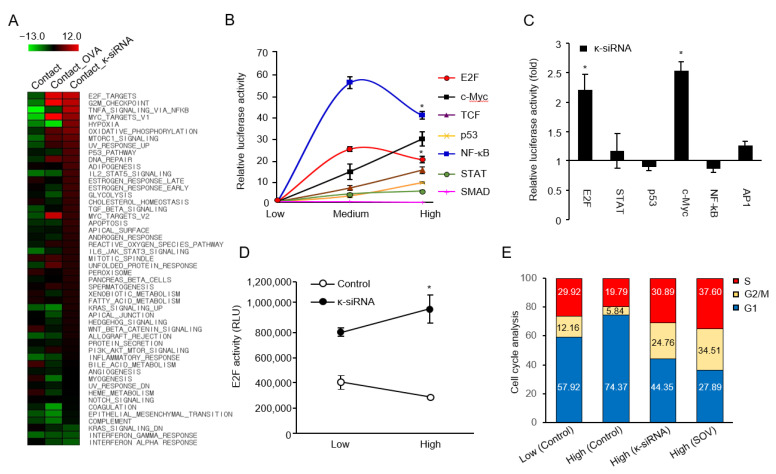
Analysis of growth-related signaling pathways in relation to cell density and R-PTP-κ expression level. (**A**) Functional enrichment analysis using 50 hallmark gene sets of gene expression profiles derived from RNA sequencing. (**B**) Activity profiles of growth-related signaling pathways under different cell-contact conditions. The activity was measured as firefly luciferase activity normalized by Renilla luciferase activity, an internal control for transfection efficiency. * *p* < 0.05. (**C**) Effect of R-PTP-κ knockdown on the activity of growth-related signaling pathway using luciferase reporter system. * *p* < 0.05. (**D**) Analysis of E2F activity in relation to plating cell density and R-PTP-κ expression. * *p* < 0.05. (**E**) Effect of R-PTP-κ on cell cycle progression. The cell cycle profile of siRNA- or SOV-treated HeLa cells was determined using flow cytometric analysis.

**Figure 4 biomedicines-10-03199-f004:**
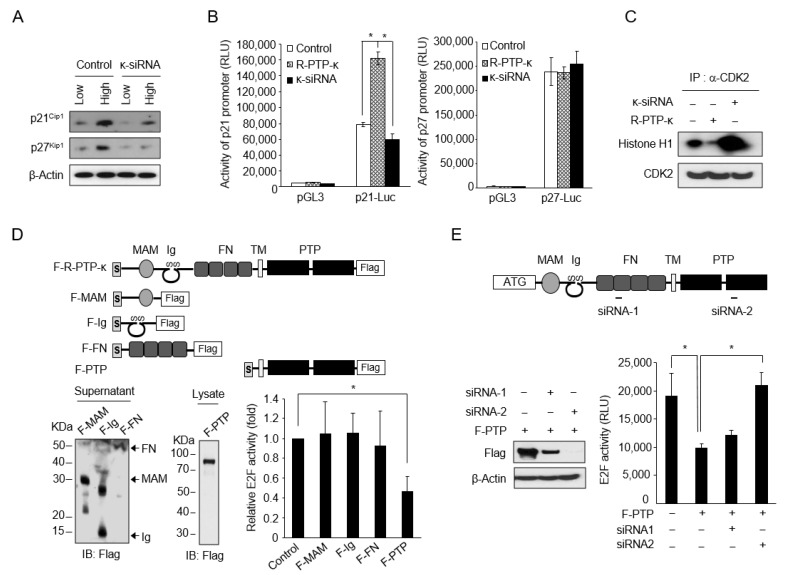
Regulation of E2F pathway by R-PTP-κ. (**A**) Analysis of p21 and p27 protein expression in relation to cell density and R-PTP-κ expression. HeLa cells transfected with control or κ-siRNA were plated at low and high density, cultured for 48 h, harvested, and analyzed for p21 and p27 protein expression by western blotting. (**B**) Effect of R-PTP-κ on the transcriptional activity of p21 and p27 in sub-confluent HeLa cells. The effect was examined by measuring firefly luciferase activity of the p21 or p27 promoter in cells overexpressing R-PTP-κ or expressing siRNA targeting R-PTP-κ. * *p* < 0.05. (**C**) Effect of R-PTP-κ on cellular CDK2 kinase activity. Total cell lysates (200 µg) were immunoprecipitated with an anti-CDK2 antibody and assayed for kinase activity using recombinant histone H1 as a substrate. The bottom panel shows the amount of CDK2 protein in each immunoprecipitate. (**D**) Effect of R-PTP-κ domains on E2F activity in HeLa cells. Schematic depiction of the structure of the various constructs cloned into a FLAG-tagged eukaryotic expression vector (upper panel). Expression of various constructs was confirmed by western blotting and E2F activity was estimated by measuring firefly luciferase activity (lower panel). * *p* < 0.05. (**E**) Regulation of E2F-dependent luciferase activity in HeLa cells by siRNA-1 and -2 targeting R-PTP-κ. HeLa cells were co-transfected with luciferase promoter-reporter plasmid, and siRNA-1 or -2 and/or the isolated PTP domain. * *p* < 0.05.

**Figure 5 biomedicines-10-03199-f005:**
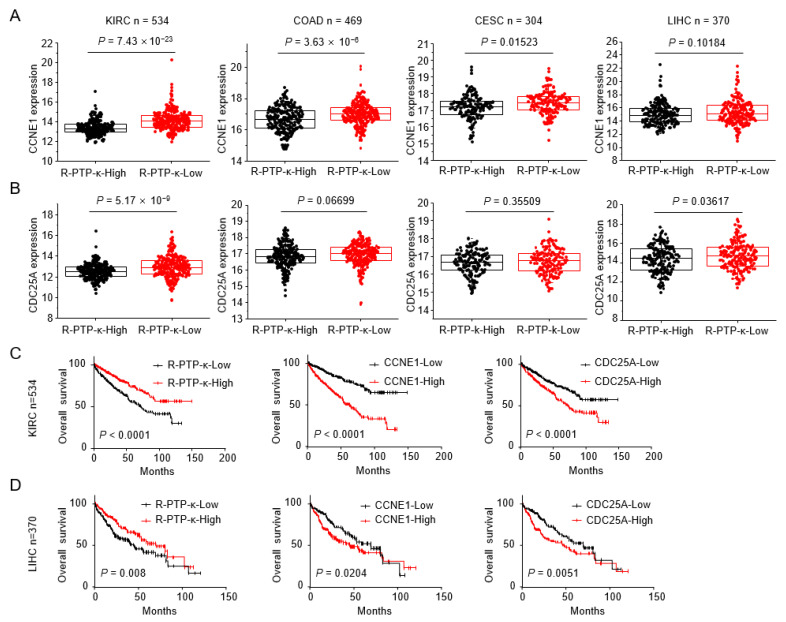
Analysis of expression correlation between R-PTP-κ and two E2F1 target genes, *CCNE1* and *CDC25A*. (**A**) Expression pattern analysis of R-PTP-κ and *CCNE1* in the public datasets generated from four different tumor types. (**B**) Expression pattern analysis of R-PTP-κ and *CDC25A* in the public datasets generated from four different tumor types. (**C**) Kaplan–Meier curve for overall survival in the KIRC cohorts according to relative expression of R-PTP-κ or the two E2F1 target genes. (**D**) Kaplan–Meier curve for overall survival in the LIHC cohorts according to relative expression of R-PTP-κ or the two E2F1 target genes.

## Data Availability

The data presented in this study are included in the Materials and Methods section. Raw RNA sequencing data were deposited in the NCBI database (GSE 212324). Further inquiries can be directed to the corresponding authors.

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
