# Peer review of "R-PTP-κ Inhibits Contact-Dependent Cell Growth by Suppressing E2F Activity"

_biomedicines, 2022, doi:10.3390/biomedicines10123199_

Round 1

Reviewer 1 Report

­This manuscript aims to demonstrate the role of R-PTP-k in contact-dependent cell growth. The authors showed that high cell density induced R-PTP-k mRNA increase, and overexpression of R-PTP-k inhibited tumorigenesis while knockdown R-PTP-k increased tumorigenesis. They further demonstrated that R-PTP-k induced increase of p21 and p27 protein level which in turn inhibited E2F activity to halt cell growth. Furthermore, patient tumor gene profiling indicated that high R-PTP-k level co-related with better survival. This manuscript addresses an important question in the cancer field and will attract a lot of interest from a broad audience.

The results are relatively clean. However, some issues need to be clarified.

(1) In Fig. 1D, the relative growth rate at both medium density and high density is around 1.0. However, in Fig. 1A, the relative growth rate at medium density is 0.4 and at high density is lower than 0.2. Can the authors explain the discrepancy? How is the relative growth rate calculated?

(2) The authors showed that R-PTP-k knockdown induced loss of contact inhibition in Fig. 1D. What happens with overexpression of R-PTP-k? Will the growth rate decrease?

(3) In Fig. 2B, the authors emphasized that the activity of E2F reporter was lower at high cell density than at medium cell density, concluding that high cell density inhibited E2F activity. However, in the same panel, the E2F reporter activity was higher at high density than at low density. Can the authors explain why that is the case? This also contradicts Fig. 4D, which showed that in control cells, E2F reporter activity was lower at high cell density than at low cell density.

(4) In Fig. 3D, the authors transfected R-PTP-k deletions to measure their effect on E2F reporter activity. It will be better to include the full-length protein in the same assay.

(5) In Fig. 4A, the p-value of CCNE1 expression was 0.10184 between R-PTP-k-High patients and R-PTP-k-Low patients in LIHC cohort. And the p-value for CDC25A was 0.35509 in CESC cohort and 0.06699 in COAD cohort. Typically, statistically significant is defined as p-value less than 0.05. The authors need to rephrase.

Author Response

(1) In Fig. 1D, the relative growth rate at both medium density and high density is around 1.0. However, in Fig. 1A, the relative growth rate at medium density is 0.4 and at high density is lower than 0.2. Can the authors explain the discrepancy? How is the relative growth rate calculated?

Response: We thank the reviewer for thorough and insightful review of our manuscript. In Fig. 1A, the relative growth rates were normalized by the number of initial cell seeding, low (1×104), medium (5×104) and high densities (4×105), to check the change of growth rate according to cell number. However, in Fig. 1D, the relative growth rate was not normalized by the number of cell seeding to determine only the effect of R-PTP-k knock-down alone on the three different cell densities. We have inserted the sentences “The relative growth rates were normalized by the number of initial cell seeding.” in the Figure1A legend.

(2) The authors showed that R-PTP-k knockdown induced loss of contact inhibition in Fig. 1D. What happens with overexpression of R-PTP-k? Will the growth rate decrease?

Response: We sincerely appreciate the reviewer’s suggestion. We investigated the cell growth rate in five stable cells overexpressing R-PTP-k. Cell growth of stable cell lines was reduced by R-PTP-k overexpression compared to control cell, pcDNA, and negatively correlated with the R-PTP-κ mRNA expression level.

(3) In Fig. 2B, the authors emphasized that the activity of E2F reporter was lower at high cell density than at medium cell density, concluding that high cell density inhibited E2F activity. However, in the same panel, the E2F reporter activity was higher at high density than at low density. Can the authors explain why that is the case? This also contradicts Fig. 4D, which showed that in control cells, E2F reporter activity was lower at high cell density than at low cell density.

Response: We thank the reviewer for the insightful comment. Like the experiment in Fig. 1A, activity of E2F reporter was normalized by the number of initial cell seeding, low (1×104), medium (5×104) and high densities (1×105), to check the change in reporter activity according to cell number. However, in Fig. 2D, the activity was not normalized by the number of cell seeding, to determine only the effect of R-PTP-k knock-down alone on the two different cell densities, We have inserted the sentences “The activity was normalized by the number of initial cell seeding.” in the Figure 3B legend.

4) In Fig. 3D, the authors transfected R-PTP-k deletions to measure their effect on E2F reporter activity. It will be better to include the full-length protein in the same assay.

Response: We thank the reviewer for the insightful comment. We had also included a full-length construct in the experiments analyzing activity of E2F reporter using R-PTP-κ structural domains. Unfortunately, we did not detect the expression of the full-length construct. The problem of the full-length R-PTP-k construct expression remains unresolved. As we wanted to determine the functional domains of R-PTP-κ that mediate contact-dependent growth inhibition, we measured the effects of R-PTP-κ structural domains on the activity of the E2F pathway.

(5) In Fig. 4A, the p-value of CCNE1 expression was 0.10184 between R-PTP-k-High patients and R-PTP-k-Low patients in LIHC cohort. And the p-value for CDC25A was 0.35509 in CESC cohort and 0.06699 in COAD cohort. Typically, statistically significant is defined as p-value less than 0.05. The authors need to rephrase.

Response: We thank the reviewer for this comment and we also agree. A p-value higher than 0.05 is not statistically significant. Therefore, we analyzed the expression correlation between R-PTP-κ and two E2F1 target genes in various cohorts to avoid statistical mistakes. Although there were cohorts with p-values higher than 0.05, the expression pattern showed a negative relationship in all cohorts. These data indicate that R-PTP-κ is inversely correlated with E2F1 target genes in various tumor tissues.

Reviewer 2 Report

I have gone through the manuscript. Topic and mechanism is indeed interesting but it needs to be presented in a better manner. R-PTP-κ has been shown to inhibit tumorigenesis. I will strongly suggest to discuss each tested cell line independently related to the effects and protein networks regulated by R-PTP-κ. Likewise, authors did not give a proper explanation about expression profiling of protein networks in tumor tissues derived from R-PTP-κ-overexpressing liver cancer cells. Detailed information about these aspects should be explained. Additionally, metastasis models should also be analyzed. How tail vein injections of R-PTP-κ-overexpressing liver cancer cells behaved in progression of metastasis in mice.  

Author Response

I have gone through the manuscript. Topic and mechanism is indeed interesting but it needs to be presented in a better manner. R-PTP-κ has been shown to inhibit tumorigenesis. I will strongly suggest to discuss each tested cell line independently related to the effects and protein networks regulated by R-PTP-κ. Likewise, authors did not give a proper explanation about expression profiling of protein networks in tumor tissues derived from R-PTP-κ-overexpressing liver cancer cells. Detailed information about these aspects should be explained. Additionally, metastasis models should also be analyzed. How tail vein injections of R-PTP-κ-overexpressing liver cancer cells behaved in progression of metastasis in mice.  

Response: We thank the reviewer for thorough and insightful review of our manuscript. Since cell contact inhibition is a critical property of normal cells to stop cell proliferation upon reaching confluence, we tested the induction of R-PTP-κ under cell-contact conditions using normal human fibroblast (HFT) cells. This property was also validated using HeLa, an immortal cell widely used in scientific research. However, these two cell lines have weak tumor-forming ability, so the effect of R-PTP-κ on tumorigenesis was examined using Chang Liver and Hep3B cells.

We are grateful for the suggestions related to expression profiling of protein networks in tumor tissues. Unfortunately, tumor tissue derived from xenograft models is not currently available. Thus, we analyzed global protein tyrosine phosphorylation status according to tyrosine phosphatase R-PTP-κ expression changes in cell lines. R-PTP-κ has been shown to dephosphorylate EGFR Tyr1068 and Tyr1173 residues, which are important for the mitogenic signaling of EGF. Our data is consistent with previous results and indicates that R-PTP-κ acts as a modulator for removing EGFR mitotic signals at high cell densities. We have included this result on page 5 of the text and in Figure 2A.

Per the reviewers's comments, it is important to elucidate the role of R-PTP-κ in progression of metastasis, but in this study, we would like to focus on its function in cell-cell contact inhibitory signaling. 

Reviewer 3 Report

The authors present a work describing that the R-PTP-k-E2F axis plays a crucial role in cell growth inhibitory signaling arising from cell-cell contact conditions.

In my opinion some crucial points need to be addressed/discussed in this study.

My comments are the following:

-        As indicated in the introduction paragraph, growth signal induced by EGF of FGF are suppressed under cell-cell contact conditions: do the authors search for these growth factors in the in vivo model proposed (i.e serum levels detection)?

A major point is to evaluate if the cells that the authors described blocked in G1 phase show features of cellular senescence: indeed, the expression of cyclin p27 and p21 are highly suggestive in this sense. This could be important since the role of senescence in tumor development and progression is still a matter of discussion as a beneficial or detrimental mechanism. 

-      The authors claim that TGFbeta1 treatment under cell-cell contact condition induces expression of p21 and p27 cyclins: TGFbeta treatment is also able to induce cellular senescence. What can be the role of TGFbeta patway in R-PTP-K mediated growth inhibition? The authors might suggest a hypothesis.

-        The authors find in the Luciferase assay, evaluating the R-PTP-k knockdown on the activity of growth-related signaling pathways, a significant increase in E2F and c-Myc pathways. These two pathways are equally involved in cell growth (Review Cell Cycle 2003 2(4):333-8. E2F1 and c-Myc in cell growth and death). Moreover, cMyc is a master regulator of ribosome biogenesis whose role in cancer is known: the authors are encouraged to deeply discuss this result taking into consideration these aspects.

In line with this, both pathways mediate apoptosis: I think that also apoptosis is needed to be evaluated in the experiments performed. The authors can also discuss the down regulation of p53 pathway since remarkably, the Rb-E2f and MDM2-p53 pathways are both defective in most human tumors having a crucial role in regulating cell cycle progression and viability ( doi: 10.1038/nrc2718)

-        The authors show that R-PTP_K suppresses anchorage-independent cell growth. Mori et al in Oncogene 2009 (DOI: 10.1038/onc.2009.139)describes an anchorage independent signature that predicts metastatic potential.  Do the authors have looked at this signature?

The role of anchorage-independent cell growth in metastasis is under debate (Zhong Deng et al, Cell Death & Disease , 2021). Do the Chang cells used in a model of  splenic injection lead to metastasis formation?

Multiple signaling pathways, including integrin transduction and its downstream signaling pathways, such as paxillin/ p130CAS, Ras-ERK, PI3K/AKT, Rho/ROCK, and YAP/TAZ pathway, are activated during detachment and contribute to anchorage-independent survival. The authors are encouraged to verify if with the model of Chang liver cells some of these pathways are activated. Moreover, they can search for ROS production in the in vivo model.

Author Response

As indicated in the introduction paragraph, growth signal induced by EGF of FGF are suppressed under cell-cell contact conditions: do the authors search for these growth factors in the in vivo model proposed (i.e serum levels detection)?

Response: We deeply appreciate the reviewer’s suggestion for improving our manuscript. To examine the function of R-PTP-κ in growth signal induced by EGF under cell-cell contact conditions, we checked EGF-EGFR mitotic signals. In the cells stimulated with EGF, overexpression of R-PTP-κ resulted in dephosphorylation of EGFR at Tyr-1173 (left in Figure), and the p-Tyr-1173-EGFR level was inversely correlated with the cell density, which substantially increased upon depletion of R-PTP-κ expression (right in Figure). This data indicates that R-PTP-κ acts as a modulator for mitogenic signaling of EGF. The new data was included in Figure 2B and C. In RNA sequencing data analysis, RNA levels of EGF and FGF were not changed in R-PTP-κ-silenced cells compared to those in control cells. Therefore, R-PTP-κ is unlikely to be involved in upstream signals regulating EGF levels, as it modulates downstream signaling of EGF.

A major point is to evaluate if the cells that the authors described blocked in G1 phase show features of cellular senescence: indeed, the expression of cyclin p27 and p21 are highly suggestive in this sense. This could be important since the role of senescence in tumor development and progression is still a matter of discussion as a beneficial or detrimental mechanism. 

Response: We also agree with the reviewer’s comment. We have inserted the sentences “Cellular senescence, a process of cell growth arrest, is a physiological mechanism of tumor suppression that inhibits the progression from benign tumor lesions to malignant tumors [40]. In our data, expression of p21, and p27, which are senescence-associated markers, was directly regulated by R-PTP-κ under cell-cell contact conditions.” in the discussion section to explain the roles of R-PTP-κ as a tumor tumor-suppressor gene.

The authors claim that TGFbeta1 treatment under cell-cell contact condition induces expression of p21 and p27 cyclins: TGFbeta treatment is also able to induce cellular senescence. What can be the role of TGFbeta patway in R-PTP-K mediated growth inhibition? The authors might suggest a hypothesis.

Response: We thank the reviewer for the valuable comment. We investigated the role of R-PTP-κ in the TGF-beta-mediated induction of p21 and p27 protein. Firstly, we performed RT-PCR analysis to determine the expression pattern of R-PTP-κ in TGF-beta treatment condition. Expression levels of R-PTP-κ dramatically increased because of TGF-beta treatment in various cell lines. In addition, TGF-beta-mediated induction of p21 and p27 proteins was abolished by R-PTP-κ knock-down. These results suggest that R-PTP-κ-mediated growth inhibition is tightly linked to the up-regulation of p21 and p27 via TGF-beta treatment.

The authors find in the Luciferase assay, evaluating the R-PTP-k knockdown on the activity of growth-related signaling pathways, a significant increase in E2F and c-Myc pathways. These two pathways are equally involved in cell growth (Review Cell Cycle 2003 2(4):333-8. E2F1 and c-Myc in cell growth and death). Moreover, cMyc is a master regulator of ribosome biogenesis whose role in cancer is known: the authors are encouraged to deeply discuss this result taking into consideration these aspects.

Response: We thank reviewer for the insightful comment. MYC has been described as a global regulator of various cellular processes, including transcription, regulation of chromatin structure, translation, DNA replication and, most recently, ribosome biogenesis. MYC is essential to the latter’s physiological functions as well as its pathological role in tumorigenesis. In addition, the perturbation of ribosome biogenesis leads to cell cycle arrest and/or apoptosis, while upregulated ribosome production promotes neoplastic transformation. Therefore, the functional study of R-PTP-k in the ribosome biogenesis pathway is a very interesting topic. However, in this study, we would like to focus on role of R-PTP-k in E2F-mediated cell contact inhibition. 

In line with this, both pathways mediate apoptosis: I think that also apoptosis is needed to be evaluated in the experiments performed. The authors can also discuss the down regulation of p53 pathway since remarkably, the Rb-E2f and MDM2-p53 pathways are both defective in most human tumors having a crucial role in regulating cell cycle progression and viability ( doi: 10.1038/nrc2718)

Response: We thank the reviewer for the valuable comment. Therefore, to determine whether R-PTP-κ expression was implicated with cell apoptosis, we performed apoptosis-related experiments. Expression of apoptosis-related protein, cleaved-PARP, was not changed by R-PTP-κ knock-down in high-cell density. This result requires further discussion with the role of R-PTP-κ in the apoptosis pathway proposed by a previous study (Sun et al., Int J Oncol. 2013 43, 1560-1568. doi: https://doi.org/10.3892/ijo.2013.2082). Sun et al. described that R PTP-κ negatively regulates the apoptosis of prostate cancer cells via the JNK pathway. Nevertheless, in this study, R-PTP-κ plays an important role in cell contact inhibition through E2F-mediated cell cycle control. In our data, the activity of p53-responsive reporter was not altered in response to changes in R-PTP-κ levels (Figure 3C). Therefore, a further study on MDM2-p53 pathway was not performed in this study.

The authors show that R-PTP-K suppresses anchorage-independent cell growth. Mori et al in Oncogene 2009 (DOI: 10.1038/onc.2009.139)describes an anchorage independent signature that predicts metastatic potential.  Do the authors have looked at this signature?

Response: We thank reviewer for the valuable comment. The MYC pathway is directly involved in the cellular process to acquire anchorage-independence. We examined the expression pattern of myc-interacting proteins regulating transcription of c-myc target genes. The expressions of c-myc repressor genes, Bin1 and Miz, were induced by overexpression of R-PTP-κ and reduced in the R-PTP-κ-silenced cells by κ-siRNA. The expression of a positive regulator, Max, was also induced by κ-siRNA. In addition, the protein level of Bin1 was increased by overexpression of R-PTP-κ and reduced by R-PTP-κ knock-down. Therefore, R-PTP-κ modulates the transcriptional activity of c-myc, which is included in anchorage independent signature.

The role of anchorage-independent cell growth in metastasis is under debate (Zhong Deng et al, Cell Death & Disease , 2021). Do the Chang cells used in a model of  splenic injection lead to metastasis formation?

Response: When evaluating the metastatic ability of liver cancer cells using a spleen injection model, Chang Liver cells do not colonize the liver, while Huh-7 cells cause metastasis. 

Multiple signaling pathways, including integrin transduction and its downstream signaling pathways, such as paxillin/ p130CAS, Ras-ERK, PI3K/AKT, Rho/ROCK, and YAP/TAZ pathway, are activated during detachment and contribute to anchorage-independent survival. The authors are encouraged to verify if with the model of Chang liver cells some of these pathways are activated. Moreover, they can search for ROS production in the in vivo model.

Response: We thank reviewer for the insightful comment. Previous studies have shown that R-PTP-κ functions as a key mediator of the cell adhesive pathway, including paxillin/ p130CAS, YAP/TAZ and Rho/ROCK (Young, K. A. et al., Biochem. J. 2021, 478, 1061–1083. doi: https://doi.org/10.1042/BCJ20200511). R-PTP-κ also regulates AKT signaling and has a critical role in colon cancer progression (Shimozato, O. et al, Oncogene 2015, 34, 1949-1960. doi: https://doi.org/10.1038/onc.2014.141). With the exception of these pathways, we determined the role of R-PTP-κ in regulation of ERK activity in Chang Liver cells because EGF-ERK mitotic signaling is critical for regulating cell growth. ERK activity was dramatically reduced by R-PTP-κ overexpression. The opposite pattern occurred to R-PTP-κ-silenced cells. Thus, our data indicate that R-PTP-κ is closely related to the regulation of ERK pathway facilitating cell growth signaling. Unfortunately, tumor tissue derived from xenograft models is not currently available. Therefore, we cannot confirm ROS production in the in vivo model.

Round 2

Reviewer 1 Report

I don't think my concern about the growth rate and E2F activity was sufficiently addressed. In fact, if cell number is used to normalize the E2F activity, a low density should have even higher adjusted activity than before, which should be opposite from the new Fig.3B. The authors should use the same analysis for Fig. 1A and 1D, and for the new Fig.3B and 3D. 

For the significance of the cohort results, the authors need to rephrase to only refer statistically significant cohorts as "significant", not the other ones. 

Author Response

I don't think my concern about the growth rate and E2F activity was sufficiently addressed. In fact, if cell number is used to normalize the E2F activity, a low density should have even higher adjusted activity than before, which should be opposite from the new Fig.3B. The authors should use the same analysis for Fig. 1A and 1D, and for the new Fig.3B and 3D.

Response: We thank the reviewer for insightful review of our manuscript and apologize for any confusion in our first response. In Fig. 1A, the relative growth rates were normalized by the initial cell seeding number to evaluate the changes in growth rate based on initial cell number. However, as mentioned in the Materials and Methods, the luciferase activity of the E2F reporter in Figures 3B and 3D was normalized to the activity of Renilla luciferase, which was used as an internal control to determine the amount of reporter construct which was successfully transfected into the cells. The two experiments were analyzed using different normalization methods to reduce experimental errors. Therefore, we have deleted the sentence “The activity was normalized by the number of initial cell seeding” in the legend of Figure 3B.

For the significance of the cohort results, the authors need to rephrase to only refer statistically significant cohorts as "significant", not the other ones.

Response: We thank the reviewer for thorough and insightful review of our manuscript. We have deleted “significant” from the results section about Figures 5A and B on page 8 of the text as suggested, so that only statistically significant cohorts are deemed “significant”.

Reviewer 2 Report

Looks in good form now. 

Author Response

Looks in good form now.

Response: We deeply appreciate the reviewer’s approval of our manuscript.

Reviewer 3 Report

I appreciate the work done by authors to answer my questions but I still have some concerns:

Regarding the senescence phenotype I was asking if the cells that authors treated (e.g. Hela cells) and found express high p21 and p27 levels are senescent. Do they express B-gal? The authors might stain cells with this marker that is the universal accepted marker for cellular senescence.

Authors show metastatic spreading of HU-7 cells that is not present with Chang cells. Do Chang cells used for in vivo splenic injections stable express R-PTP-K as in the other in vivo model in which the control was done with mock vector? In the in vitro model of anchorage independent cell growth authors use Hep3b cells, do these cells show same result in terms of metastasis formation? Why authors didn't inject Hep3b cells? Probably I miss the point, but I didn’t understand the differential expression of R-PTP-K expression in cell lines used in vivo. In my opinion the presence/absence of metastatic spreading by cells which have differential expression of R-PTP-k can support its role as tumor suppressor. In this case I think that this datum needs to be presented in detail in the paper.

Regarding EGF/FGF analysis, I was asking if the authors can test these growth factors in sera of xenograft mice. The same will be done for ROS measurement even in the absence of tissues. 

Author Response

Regarding the senescence phenotype I was asking if the cells that authors treated (e.g. Hela cells) and found express high p21 and p27 levels are senescent. Do they express B-gal? The authors might stain cells with this marker that is the universal accepted marker for cellular senescence.

Response: We greatly appreciate your thoughtful comments. We performed β-galactosidase (β-gal) staining to detect senescence of HeLa cells under cell-contact conditions. Senescence-associated β-gal staining did not increase significantly under cell contact conditions. These results are presumably because β-gal is present only in senescent cells, and not found in pre-senescent, quiescent, or immortal cells. Therefore, it is difficult to accurately describe the relationship between contact inhibition of HeLa cells and senescence in this study. We have deleted the sentences “Cellular senescence, a process of cell growth arrest, is a physiological mechanism of tumor suppression that inhibits the progression from benign tumor lesions to malignant tumors [40]. In our data, expression of p21, and p27, which are senescence-associated markers, was directly regulated by R-PTP-κ under cell-cell contact conditions.” in the discussion section of revised manuscript.

Authors show metastatic spreading of HU-7 cells that is not present with Chang cells. Do Chang cells used for in vivo splenic injections stable express R-PTP-K as in the other in vivo model in which the control was done with mock vector? In the in vitro model of anchorage independent cell growth authors use Hep3b cells, do these cells show same result in terms of metastasis formation? Why authors didn't inject Hep3b cells? Probably I miss the point, but I didn’t understand the differential expression of R-PTP-K expression in cell lines used in vivo. In my opinion the presence/absence of metastatic spreading by cells which have differential expression of R-PTP-k can support its role as tumor suppressor. In this case I think that this datum needs to be presented in detail in the paper.

Response: We thank the reviewer for their detailed comments. We performed in vivo spleen injections using unestablished parental cell lines to evaluate the metastatic potential of various liver cancer cells including Chang Liver and Hep3B, because not all immortalized cancer cells have invasive and metastatic capacity. Consistent with previous studies, Hep3B and Chang Liver cell micrometastases were not observed in the liver or other organs (Liu et al., Oncology Letters 2018, 16, 335-345. doi: https://doi.org/10.3892/ol.2018.8666). Anchorage-independent growth is a hallmark of anoikis resistance and tumor metastasis, but is also closely related to tumor initiation capacity (Borowicz et al., J. Vis. Exp. 2014; 92, 51998. doi: 10.3791/51998). In addition, cell contact initiates cell cycle arrest and downregulates proliferation and mitogen signaling pathways, regardless of external factors. Therefore, to confirm whether R-PTP-k has tumor suppressing activity in vivo, we investigated its inhibitory effect on tumor growth in Chang Liver-derived xenograft mouse models. We agree with the reviewer that it is important to elucidate the role of R-PTP-κ in the process of tumor metastasis, but in this study we wanted to focus on the molecular mechanisms of cell contact-mediated growth control. As shown in the figure below, the expression level of endogenous R-PTP-κ was relatively high in Hep3B but low in Chang Liver cells. We have modified the sentence from “ Conversely, the shRNA-mediated knockdown of R-PTP-κ in Hep3B cells resulted in enhanced colony formation.” to “Conversely, the shRNA-mediated knockdown of R-PTP-κ in Hep3B cells, in which endogenous R-PTP-κ level is relatively high, resulted in enhanced colony formation.” in the results section of revised manuscript.

Regarding EGF/FGF analysis, I was asking if the authors can test these growth factors in sera of xenograft mice. The same will be done for ROS measurement even in the absence of tissues.

Response: We thank the reviewer for the comment and apologize for our insufficient first response. We measured tumor growth and volume at the time of sacrifice, but did not obtain sera from the xenografted mice. Unfortunately, it is currently difficult to establish new xenograft mouse models to obtain sera and perform the suggested experiments. However, according to the reviewer's comment, we investigated changes in ROS production due to cell contact in cellular systems. ROS detection was performed using the DCFDA assay kit. ROS production increased in cell contact conditions but was not affected by changes in R-PTP-k expression. Therefore, it is difficult to accurately describe the relationship between cell contact-induced production of ROS and R-PTP-k in this study. 

Round 3

Reviewer 1 Report

The authors addressed my questions.